# A pH-Responsive Asymmetric Microfluidic/Chitosan Device for Drug Release in Infective Bone Defect Treatment

**DOI:** 10.3390/ijms24054616

**Published:** 2023-02-27

**Authors:** Hongyu Chen, Wei Tan, Tianyi Tong, Xin Shi, Shiqing Ma, Guorui Zhu

**Affiliations:** 1School of Chemical Engineering and Technology, Tianjin University, Tianjin 300350, China; 2School and Hospital of Stomatology, Tianjin Medical University, Tianjin 300070, China; 3Department of Stomatology, The Second Hospital of Tianjin Medical University, Tianjin 300211, China

**Keywords:** microfluidic device, pH-responsive drug release, guided bone regeneration

## Abstract

Bacterial infection is currently considered to be one of the major reasons that leads to the failure of guided bone regeneration (GBR) therapy. Under the normal condition, the pH is neutral, while the microenvironment will become acid at the sites of infection. Here, we present an asymmetric microfluidic/chitosan device that can achieve pH-responsive drug release to treat bacterial infection and promote osteoblast proliferation at the same time. On-demand release of minocycline relies on a pH-sensitive hydrogel actuator, which swells significantly when exposed to the acid pH of an infected region. The PDMAEMA hydrogel had pronounced pH-sensitive properties, and a large volume transition occurred at pH 5 and 6. Over 12 h, the device enabled minocycline solution flowrates of 0.51–1.63 µg/h and 0.44–1.13 µg/h at pH 5 and 6, respectively. The asymmetric microfluidic/chitosan device exhibited excellent capabilities for inhibiting *Staphylococcus aureus* and *Streptococcus mutans* growth within 24 h. It had no negative effect on proliferation and morphology of L929 fibroblasts and MC3T3-E1 osteoblasts, which indicates good cytocompatibility. Therefore, such a pH-responsive drug release asymmetric microfluidic/chitosan device could be a promising therapeutic approach in the treatment of infective bone defects.

## 1. Introduction

Bone defects caused by trauma, severe infection or tumors are a critical problem for many patients who need surgery and remain a major challenge in clinical practice [1]. Nowadays, guided bone regeneration (GBR) technology is generally accepted as a therapeutic modality in the treatment of periodontal disease or bone defects in other parts of the body [2]. As a key medical device of GBR procedures, GBR membranes (GBRM) can prevent cell migration from connective tissue and epithelium into bone defect sites, thereby providing space for osteoblast cell growth and achieving bone regeneration [3,4]. Owing to their good biocompatibility and low toxicity, polytetrafluoroethylene (PTFE) membranes, collagen membranes and artificial or natural polymer membranes are the main commercial membranes currently used in clinics [5,6]. Even though current commercial GBRMs are well tolerated by the human body, they suffer from prospective complications like implant-associated infection, resulting in implant failure along with delayed healing [7,8,9]. The implanting surgery of GBR may cause bacterial infection, and severe bacterial infection then leads to the failure of GBR therapy, especially for the application in the oral environment [8,10]. Consequently, bacterial infection may lead to secondary surgery, which could increase not only patient suffering but also health costs. Such problems motivate researchers to develop functional GBRMs with excellent antibacterial functions that can fight infection for better bone regeneration [11].

Most previous works have been devoted to the preparation of GBRMs loaded with antibiotics through physical or chemical approaches [12]. Xue et al. [13] developed a GBR membrane by electrospinning poly(ε-caprolactone) (PCL) and gelatin blended with metronidazole (MNA), and the antibiotic thereof was used to prevent infection. In addition, GBR membranes with antibacterial coatings have been developed [14]. Nardo et al. [15] improved the antibacterial effects of a PTFE membrane by coating AgNO_3_ on the surface of the membrane. In fact, direct doping of antibiotics into the membrane or making antibacterial coatings are a simple approach, but these strategies can lead to the spontaneous nondependent release of drugs [16]. Delivery of antibiotics by external stimulations including temperature, ultrasound or magnetic fields have been developed [17,18]. For instance, Sirinrath et al. [19] embedded ciprofloxacin (CIP) and iron oxide magnetic nanoparticles (NPs) in PCL microspheres. Through the action of an alternating magnetic field, antibiotic CIP was released from the microspheres. These approaches can achieve antibiotic release at a specified location and time, but they do not have an intrinsic ability to respond to the bacterial-infection-induced local microenvironment, and they have to rely on external stimulations [20]. Some approaches have been developed that do not release antibiotics in infection-free environments but furnish antibiotics in acid circumstances caused by bacterial infection [21]. Under normal physiological conditions, the pH is 7.4. Bacterial infection is associated with acidity and the pH decreases to 4.5–6.5 at the site of infection [22,23]. Such drug-delivery systems use pH-responsive polymers/hydrogels to entrap the desired drug into their polymeric network or pH-responsive acid-labile linkers to incorporate drugs in biomaterials. One scenario is that these pH-responsive polymers/hydrogels [24] undergo swelling or degradation in acid environments, thereby allowing the diffusion of therapeutic agents [25]. Another scenario is that, under the action of the acid cleavable linker, drugs in biomaterials can be released [26]. 

Although many controlled drug-delivery systems have shown potential, their ability of precise control needs to be further strengthened. In recent years, the microfluidic technique has garnered increasing attention in the field of drug release due to its precise control of fluids [27,28,29]. Lee et al. [30] proposed a microdevice concept for ocular drug delivery, where the device consisted of reservoirs and microchannels. Drug solutions diffused in the microchannel could achieve a stable flowrate, and different flowrates were controlled by changing microchannel configurations. For the control of activation time of drug release, Yang et al. [31] developed a microfluidic device that had release channels filled with polymer poly (DL-lactide-coglycolide). The degradation of polymers delayed the pace of drug release. Indeed, biodegradable polymers with different degradation speeds can modulate the starting time of drug release. Some non-mechanical micropumps have been used for drug delivery, without the need for external stimulation. Drug release can be actuated by environmental changes [32,33,34]. Kim et al. [35] developed a leaf-inspired hydrogel micropump, which consisted of a thermo-sensitive porous membrane and agarose gel. The membrane absorbed water from the reservoir and delivered it to the agarose gel via an evaporation-induced water potential difference. Since the shrinking and swelling behaviors of the membrane could be manipulated by temperature variation, this responsiveness of the membrane enabled the control of evaporation and pumping rates. 

Antibiotic-loaded implants giving spontaneous and nondependent drug release with an inappropriate dose may increase the risk of antimicrobial resistance [36]. Therefore, it is necessary to develop novel systems that can store antibiotics in infection-free environments and release them in acid circumstances caused by bacterial infection. Furthermore, it is also necessary to control the drug release process to avoid high-dose-induced adverse side effects. Herein, we introduce the microfluidic technique into GBR implants, and put forward a novel device that can achieve pH-responsive drug release and has a more controlled drug release process.

In this work, we propose an asymmetric microfluidic/chitosan device that is ready for drug loading, can be manufactured by simple methods, and can achieve pH-responsive drug release without the need for external stimulation. As shown in Figure 1, we hypothesize that the chitosan membrane side is beneficial for osteoblast adhesion and proliferation, whereas the microfluidic side can prevent fibrous tissue infiltrating into the bone defect region and realize pH-responsive drug release for bacterial infection treatment at the same time. The microfluidic side is a pH-regulated drug-delivery micropump that utilizes a pH-sensitive hydrogel as an actuator to dispense antibiotics. The micropump has a polymeric chamber that serves as the drug reservoir with a hydrogel inside. A thin elastic membrane separates the hydrogel and the drug solution, and deflects when the hydrogel starts to swell, thus forcing the drug out of the reservoir. The pH-sensitive hydrogel has a higher swelling ratio in acid environments than in neutral conditions, thus different drug release performances occur under different pH values. Herein, the antibiotic minocycline was chosen due to its broad-spectrum antibacterial properties. The antibacterial performance of the microfluidic/chitosan device was tested against *Staphylococcus aureus* (*S. aureus*) and *Streptococcus mutans* (*S. mutans*). In addition, the assessments of cytocompatibility of the microfluidic/chitosan device were examined using L929 fibroblasts and MC3T3-E1 osteoblasts. 

## 2. Results and Discussion

### 2.1. pH-Sensitive Hydrogel Swelling Kinetics

The swelling ratio of the hydrogel at three different pH values is shown in Figure 2. It is obvious that the PDMAEMA hydrogel has pronounced pH-sensitive properties. The swelling ratio decreases with the increase of the pH, and in the acid environment the swelling ratio is much higher than that in the neutral one. This is because PDMAEMA hydrogels are cationic polymers. In acid conditions, the tertiary amine groups in the PDMAEMA polymer chains undergo protonation thus creating cationic charge within the polymer. The electrostatic repulsion between the polymeric chains prompts them to move away from each other [37,38], and induces an increasing of the volume and the swelling rate of the gel in low pH environments. In the neutral environment, the tertiary amine groups have a low degree of protonation, thus resulting in a low swelling ratio. As shown in Figure 2, as time went by, the swelling ratio of the hydrogel increased over 24 h. There was no significant difference in the hydrogel swelling ratio between 24 h and 30 h at all three pH conditions, which indicates that the hydrogel reached an equilibrium state within 24 h. The equilibrium swelling ratio of the hydrogel was 15.61, 11.76 and 3.61 at pH 5, 6 and 7.4, respectively. At pH 7.4, the hydrogel showed a slight volume transition during the whole swelling process, and at pH 5 and 6 the swelling ratio of the hydrogel increased significantly. The hydrogel also showed a faster swelling rate in the early stage, especially in the first 1 h. In the pH 5 condition, the swelling ratio was 4.72 in 1 h, and was 3.48 in the pH 6 condition. Figure 2c shows the vertical view of three hydrogel bars (same initial size) when they reached the equilibrium state at different pH conditions. These results reveal that in acid pH conditions (bacterial infection regions) the hydrogel undergoes a large volume change, which is higher than that in non-infection regions (neutral pH). Thus, the PDMAEMA hydrogel is regarded as an actuator to trigger drug release when bacterial infection occurs.

### 2.2. Drug Release Performance

The quantitative characterization of drug-delivery capacity of the asymmetric microfluidic/chitosan device is presented in Figure 3. Minocycline release was driven by volume transition of the hydrogel and we monitored the release process within 12 h. Overall, there was no minocycline release at pH 7.4 due to the low swelling ratio of the hydrogel, while the release of antibiotic thereof only occurred in acid environments, which indicates that selective drug release can be achieved under the control of hydrogel. Under the pH 5 condition, hydrogel exhibited a higher swelling ratio and thus could lead to a higher release amount than under the pH 6 condition. The cumulative release amount reached 14.94 µg and 12.17 µg in12 h, respectively. In acid environments, faster flowrates were found in the first 1 h, as a result of the rapid increase of the hydrogel swelling rate. The average flowrate in the first 1 h at pH 5 was 5.97 µg/h, and 3.99 µg/h at pH 6. Although there was an initial burst release of minocycline, in the following period the flowrate varied slightly between 0.51 and 1.63 µg/h and 0.44 and 1.13 µg/h at pH 5 and 6 conditions, respectively. In essence, the flowrate can be controlled by the volume transition rate. If using hydrogels with different volumes or crosslinking degrees, flowrates over a period can change because of the distinguishing volume transition rates. 

We also assessed the ‘on–off’ characteristics of the device to verify its sensitivity to pH changes. First, in order to study the swelling and de-swelling performance of the hydrogel in a low–high pH cycle, we put the hydrogel over pH 5 buffer solution for 2 h then transferred it to a pH 7.4 condition for 1 h, and the volume of hydrogel was monitored up to 8 h. As shown in Figure 4a, when the hydrogel was transferred from pH 5 to 7.4, the swelling ratio decreased. This can be explained by the fact that, at the pH 7.4 condition, the tertiary amine groups of the hydrogel had a low degree of protonation, thereby the electrostatic repulsion between polymer chains decreased and the hydrophilicity of chains decreased at the same time. When the hydrogel was in contact with a pH 5 environment, the protonation degree increased and it started to swell again. The device was also subjected to the aforementioned low–high pH cycle. The device released 6.49 µg at pH 5 over 2 h, and at pH 7.4 this amount remained for over 1 h (see Figure 4b). The next cycle illustrated a similar pattern, and the cumulative release amount was 10.12 µg at the end of 8 h. According to the results, in non-infection or healed regions, no drug was released from the device, while, with the occurrence of bacterial infection, the device started releasing minocycline and, under the control of a hydrogel actuator, could achieve a relatively stable flowrate. 

### 2.3. In Vitro Antibacterial Performance

Implant-related infection can be caused by different bacterial species. Minocycline was chosen as an appropriate antibiotic in our experiments due to its broad-spectrum antibacterial ability [39]. We used *Staphylococcus aureus*, the common bacteria at the infection site, and *Streptococcus mutans*, which only grows in oral environments to validate the antibacterial properties of the asymmetric microfluidic/chitosan device. Minocycline release solution was obtained at certain times (2, 6, 12, 24 h) from the pH 5 and 6 buffer. The bacteria were cultured in a mixture of the minocycline solution and medium with a volume ratio of 1:1. After 24 h of cocultivation, the OD value of each group was measured. Similarly, buffer release solutions (pH 5 and 6) from blank samples were cocultured with the bacteria in a 1:1 (*v*/*v*) ratio. The buffer solution containing minocycline was replaced by phosphate buffer saline (PBS) in the control group. Figure 5a,b show that the OD values of each group decreased to a low level for the minocycline release solution coculture with *S. aureus* and *S. mutans*, in comparison with the buffer release solution from the blank samples. It was also found that when bacteria were cultured in an acid environment their growth could be inhibited to some extent, and the results indicated that *S. aureus* and *S. mutans* were both less acid tolerant. 

### 2.4. In Vitro Cytocompatibility

In order to study the effect of minocycline concentration on L929 fibroblast growth, we prepared a minocycline-containing medium whose concentration was equivalent to when the asymmetric microfluidic/chitosan device released at pH 5 for 24 h. As shown in Figure 6a, the CCK-8 testing results suggest that the cells proliferated continuously in both groups and grew rapidly after one day. The OD values of these two groups were similar on the first day, then, according to the data from the third and fifth day, cells that were cultured with the minocycline-containing medium showed a better proliferation trend compared with the control group. Some previous studies have showed that an appropriate amount of minocycline can promote the proliferation of periodontal ligament fibroblasts [40]. We further performed an AO/EB dyeing experiment for 1, 3, and 5 days, and the results in Figure 6b show that cells cultured with minocycline-containing medium proliferated normally in 5 days. Fibroblasts were also seeded on the surface of the micropump to study the effect of PLA materials on cell growth. It can be seen that the cells on the surface of the micropump spread normally over 5 days and were in spindle morphology (Figure 6b). 

The loose chitosan membrane of the asymmetric microfluidic/chitosan device was used to promote osteoblast adhesion and proliferation. Figure 7a shows the CCK-8 assay results of the MC3T3-E1 osteoblasts seeded in the wells and on the surface of the chitosan membrane after 1, 3, and 5 days. After 1 day, cells seeded on the surface of the chitosan membrane proliferated and underwent an increasing trend, where the OD value on the fifth day was higher than that of the control group, which indicates that the chitosan membrane promoted osteoblast proliferation. LSCM images in Figure 7b showed after 5 days of culturing that the cells in the control and chitosan conditions spread normally over 5 days. A SEM micrograph of the chitosan surface with osteoblasts, shown in Figure 7c, displayed a loose and porous morphology. After 5 days of culturing, the osteoblasts showed a round shape and were anchored to the chitosan surface. Thus, we can conclude that the asymmetric microfluidic/chitosan device had satisfactory cytocompatibility. 

In order to fight bacterial infection for achieving better bone regeneration, many studies have been devoted to the preparation of GBRMs loaded with antibiotics by doping antibiotics into the membranes through physical or chemical approaches. Drugs are released from the membrane in a spontaneous and nondependent manner. In this study, we proposed an asymmetric microfluidic/chitosan device that can achieve a pH-responsive drug release. On-demand release of minocycline relies on a hydrogel actuator. The PDMAEMA hydrogel has pronounced pH-sensitive properties, i.e., the swelling ratio of the hydrogel in acid environments is much higher than that in a neutral environment. The flowrates can be controlled by the volume transition rate of the hydrogel. Furthermore, compared with the method of doping antibiotics into membranes, our device is readily for drug loading. The drug in the drug reservoir can be easily replaced according to the requirements, which is more flexible in clinical use. In addition, the device in this study shows satisfactory antibacterial ability and cytocompatibility. The pH-responsive asymmetric microfluidic/chitosan device developed in the current study can be an efficient drug-delivery system due to its on-demand release capability. 

In this study, we introduced a microfluidic technique into GBR implants and put forward a concept of using the volume transition of the hydrogel to control the drug release rate. A future study will investigate the tissue compatibility and bone regeneration behavior of the implants in vivo.

## 3. Materials and Methods

### 3.1. Ethics Statement

All cell lines used in this paper (Mouse Fibroblast L929 and Mouse Osteoblast MC3T3-E1) were provided by Tianjin Medical University (Tianjin, China). All experiments were performed in accordance with the relevant guidelines set by the National Health Commission of the People’s Republic of China and approved by the ethics committee at Tianjin Medical University (Tianjin, China).

### 3.2. Syntheses of PDMAEMA Hydrogel

Poly [2-(dimethylamino) ethyl methacrylate] (PDMAEMA) is a kind of pH-sensitive polymer and it has been widely used in the biomedical field due to several advantageous features such as non-toxicity and biocompatibility [41,42]. The PDMAEMA hydrogel was prepared through the radical aqueous solution polymerization route. A total of 4 g 2-(dimethylamino) ethyl methacrylate (DMAEMA) (Sigma Aldrich, Shanghai, China) was passed through a basic alumina column (DIKMA, Beijing, China) before using, in order to remove its inhibitor. A DMAEMA monomer was dissolved in 16 mL de-ionized water, then 0.06 g Methylene-bis-acrylamide (MBA, cross-linking agent) (Sigma Aldrich, China) and 0.04 g potassium persulfate (KPS, initiator) (Sigma Aldrich, China) were added. The solution was mixed under a nitrogen atmosphere for 10 min to remove oxygen, then the sealed reaction tube was kept for 24 h at a 60 °C atmosphere for the completion of polymerization. After the reaction, the tube was cooled to room temperature and the plunger was opened to take out the sample, with subsequent cutting of the prepared gel into small pieces. The pieces were soaked and washed with de-ionized water, while changing the water continuously to remove the unreacted monomer, cross-linking agent and initiator. After 3 days, the gel was taken out and dried for later use.

### 3.3. Device Fabrication

The structure of the asymmetric microfluidic/chitosan device for treating bone defects is shown in Figure 8. It contains two parts, named as loose chitosan membrane side (part 1) and microfluidic side (part 2). The loose structure that contacts the bone defect space directly is beneficial for osteoblast adhesion and blood clot stabilization [43,44]. Chitosan(poly(1,4-D-glucosamine)) is a natural biopolymer and it is easily processed into membranes. Due to its biocompatibility and non-toxicity, some studies have used chitosan membrane for bone tissue regeneration [1,45,46]. As such, a loose chitosan membrane was chosen to contact with bone defect spaces. The microfluidic side is a pH-regulated drug-delivery micropump, which can release minocycline to treat possible bacterial infection. The 3D exploded view of the pH-regulated drug-delivery micropump is shown in Figure 8a. The top layer is a polylactic acid (PLA) drug reservoir with a groove structure, and the intermediate layer is a PLA elastic membrane (10 µm). Due to the biocompatibility, low cost and non-toxicity of PLA, it has been widely used in GBR technology [47,48]. Under the elastic membrane, a pH-sensitive hydrogel (4 × 4 × 1 mm^3^) is wrapped and embedded in the drug reservoir (16 mm in diameter, 2 mm height). The elastic membrane separates the minocycline solution and the hydrogel, and the minocycline solution is stored in the drug reservoir. The bottom layer is a PLA grid plate (300 µm), which is attached to the hydrogel, exposing the hydrogel to the aqueous contents while simultaneously providing mechanical protection. The micro-outlets prevent the leakage of solution without external force, and the hydrostatic pressure is insufficient to overcome surface tension. Drug outlets are distributed along the circumference, and the number of outlets can be varied according to the requirements. The micropump in this work has four drug outlets (0.8 mm in diameter) in different directions. 

Figure 9 shows the fabrication process of the asymmetric microfluidic/chitosan device. For the microfluidic side, the hydrogel was located in the center of the grid plate, then using the elastic membrane to cowl the hydrogel, the elastic membrane and the grid plate were bonded together by medical grade adhesive. Finally, the drug reservoir was stuck on the top of the elastic membrane. The loose chitosan membrane (400 µm) was on the upper layer of the drug reservoir. For loose chitosan membrane synthesis, 2 g chitosan powder (Sigma Aldrich, China) was dissolved in 2% acetic acid solution to prepare a 2% (*w*/*v*) chitosan solution, and the mixed solution was stirred evenly with a magnetic stirrer. Then, the chitosan solution was slowly poured on a polytetrafluoroethylene mold at room temperature to obtain even liquid films; thereafter the films were immersed in liquid nitrogen together with the mold for 10 s. Subsequently, the films were lyophilized at −80 °C for 24 h, and the loose chitosan membrane obtained after vacuum freeze-drying. The asymmetric microfluidic/chitosan device (16 mm in diameter, 2.7 mm height) was obtained by adhering the chitosan membrane on the upper layer of the drug reservoir. All components were bonded by medical grade adhesive.

### 3.4. Swelling Kinetics of the pH-Sensitive Hydrogel

In order to study the swelling behavior of the pH-sensitive hydrogel, the PDAMEMA hydrogel was cut into 4 × 4 × 1 mm^3^ and then put it on the grid plate over three different pH buffer solutions (5, 6 and 7.4) for 30 h at 37 °C atmosphere, where pH 7.4 was for normal physiological conditions, and pH 5 and 6 were in the typical pH range for an infected region. The swelling ratio is defined as the volume ratio between the swollen hydrogel at certain times and that of its initial state. All tests were repeated three times and averaged.

### 3.5. Drug Delivery Characterizations

For characterizing the drug-delivery capability of the asymmetric microfluidic/chitosan device, we put the device over a set of different constant pH buffer solutions (5, 6 and 7.4) at 37 °C for 12 h. The concentration of minocycline solution was 0.1 mg/mL, and the released solution was sampled periodically. In order to verify the ‘on–off’ performance of this device, it was subjected to a low–high pH cycle, where pH 5 and 7.4 buffers were chosen, with pH 5 representing the typical pH in infection regions, and pH 7.4 representing normal conditions. Within each cycle, the device was first allowed to release minocycline over pH 5 buffer for 2 h, and then for 1 h over pH 7.4 buffer. High performance liquid chromatography (HPLC) (Waters E-2695, Shanghai, China) was used to determine the minocycline concentration. All tests were repeated three times and averaged.

### 3.6. Antibacterial Activity of Minocycline Released at Different pH Values

The antibacterial efficiency of minocycline released from the device at different pH values was tested against *S. aureus* and *S. mutans*, as *S. aureus* is the most common bacteria in infection regions and *S. mutans* is the bacteria endemic to oral environments [49]. In order to ascertain the antibacterial activity of pH-responsive minocycline release, 1 mL minocycline solution released under different pH (5 and 6) was collected at specified times (2, 6, 12, 24 h), and then *S. aureus* or *S. mutans* suspensions (1 × 10^6^ CFU/mL) were added into the minocycline release solution in a 1:1 (*v*/*v*) ratio. Specimens were put into 6-well plates and incubated at 37 °C for 24 h. After the experiment, 100 µL of each bacterial culture solution was transferred into a 96-well plate, and the absorbance was measured at 600 nm in a microplate reader (RT-6000, Rayto, Shenzhen, China) to obtain the OD (optical density) value. The OD value indicates the optical density absorbed by the tested solution. Similarly, release buffer solutions (pH 5 and pH 6) from blank samples were cocultured with the bacteria in a 1:1 (*v*/*v*) ratio. The phosphate buffer saline (PBS) solution was used as the control group to culture with bacteria suspensions in a 1:1 (*v*/*v*) ratio. All tests were repeated more than three times.

### 3.7. Cytocompatibility Evaluation

MC3T3-E1 osteoblasts and L929 fibroblasts were used to evaluate the cytocompatibility of the asymmetric microfluidic/chitosan device. The proliferation of the cells was assessed by using a CCK-8 assay. As for the microfluidic side, in order to ascertain the effect of minocycline concentration on cells, we prepared a minocycline-containing complete growth medium (DMEM with 10% fetal bovine serum, 100 mg/mL of streptomycin, and 100 U/mL of penicillin), and the concentration of minocycline was equivalent to the amount released by the device in 24 h at pH 5. Subsequently, 5000 fibroblasts in the minocycline-containing medium were seeded into a 24-well plate. In the control group, cells in medium without minocycline were seeded into a 24-well plate. In order to study the effect of PLA material on cell growth, 5000 fibroblasts in normal medium were seeded on the surface of the micropump. As for the chitosan side, 5000 osteoblasts in medium were seeded onto the surface of the chitosan membrane and placed in a 24-well plate; cell suspensions added into wells with no samples were regarded as controls. Plates were incubated at 37 °C under 5% CO_2_ atmosphere. At days 1, 3, and 5 of culture, 100 µL of medium was transferred for CCK-8 assay, and the absorbance was measured at 450 nm in a microplate reader to obtain the OD value (RT-6000, Rayto, China).

For morphological observation, at days 1, 3, 5 of culture, cells in wells and on the surface of samples were washed with PBS three times, and the cells were stained using a Live/Dead Cell Double Staining Kit and observed using laser confocal microscopy (LSCM) (Fv-1000, Olympus, Tokyo, Japan). Furthermore, the morphologies of the adhered cells on the surface of the chitosan membrane were also observed using a scanning electron microscope (SEM) (Apreo, FEI, Brno, Czech Republic). Cells on the membrane were fixed using 2.5 vol% glutaraldehyde in 0.1 M sodium cacodylate buffer for 24 h at 4 °C, dehydrated using graded ethanol (25, 50, 75, 90 and 100%) and then dried before being sputtered with gold. All tests were repeated more than three times.

### 3.8. Statistical Analysis

The results were reported as the mean ± standard deviation. Significant differences between the two groups were determined by the Student’s *t*-test. When performing a hypothesis test in statistics, a *p*-value helps to determine the significance of the results. The *p*-value is the probability of obtaining a result at least as extreme as the one that was actually observed, given that the null hypothesis (no difference between groups) is true. A *p*-value stands for the probability that an observed difference could have occurred just by random chance. A difference at *p* < 0.05 was considered to be significant.

## 4. Conclusions

A novel asymmetric microfluidic/chitosan device was manufactured for pH-responsive drug release to treat bacterial infection. A PLA pH-regulated drug-delivery micropump was assembled with a loose chitosan membrane, and a pH-sensitive hydrogel in the micropump acted as an actuator. The swelling behavior of PDMAEMA hydrogel under different pH conditions was elucidated. The equilibrium swelling ratios were 15.61, 11.76 and 3.61 at pH 5, 6 and 7.4, respectively. This significant volume change in acid environments resulted in minocycline release only at pH 5 and 6, while the asymmetric microfluidic/chitosan device maintained a zero-delivery rate in the normal physiological environment. Within 12 h, the flowrates were stabilized between 0.51 and 1.63 µg/h and 0.44 and 1.13 µg/h at pH 5 and 6, respectively. The microfluidic/chitosan device also showed good ‘on–off’ characteristics. When the device was subjected to a low–high pH cycle, minocycline only released at pH 5 and stopped releasing at pH 7.4. Minocycline released from the device efficiently inhibited *S. aureus* and *S. mutans* growth in vitro within 24 h. Biocompatibility experiments showed that the asymmetric microfluidic/chitosan device had no negative effect on cell morphology, viability and proliferation of L929 fibroblasts and MC3T3-E1 osteoblasts. In short, our findings suggest that this antibacterial asymmetric microfluidic/chitosan device could be a potent therapeutic approach to control implant infection. This device can use hydrogels with different volumes and crosslinking degrees, since the change in the hydrogel’s volume transition rates could result in different drug release rates. It can also incorporate other smart hydrogels, responsive to different stimuli (specific ions, glucose, etc.) in order to achieve a broader application in the biomedical field.

## Figures and Tables

**Figure 1 ijms-24-04616-f001:**
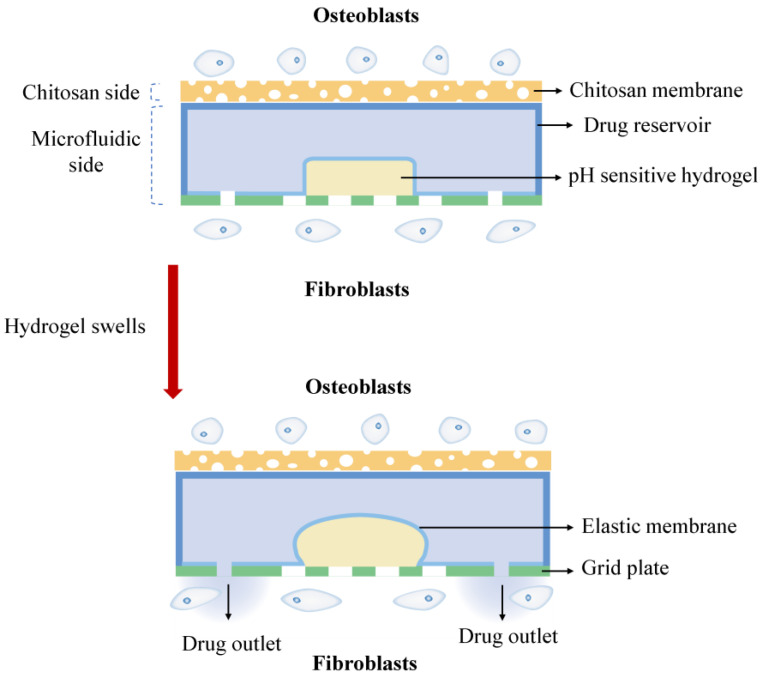
The working principle of the asymmetric microfluidic/chitosan device; the chitosan side is beneficial for osteoblast proliferation and adhesion, and the microfluidic side can release minocycline to treat bacterial infection.

**Figure 2 ijms-24-04616-f002:**
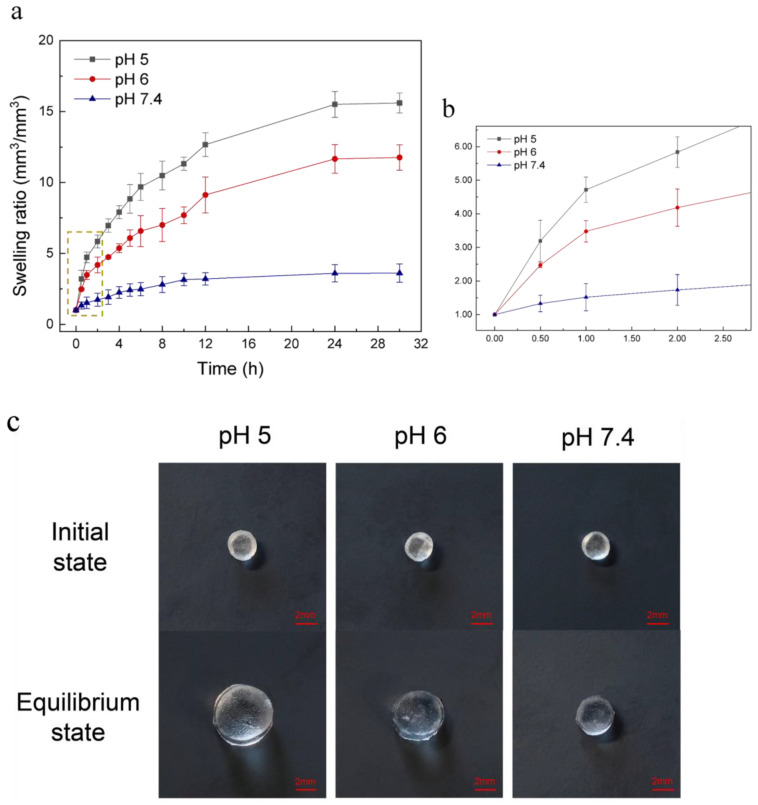
(**a**) Characterization of the pH sensitive hydrogel swelling at pH 5, 6 and 7.4 conditions over 30 h. PDMAEMA hydrogel showed pronounced pH-sensitive properties. It swelled faster in acid environments and the equilibrium swelling ratio (24 h) was 15.61 at pH 5 and 11.76 at pH 6, while the swelling ratio was only 3.61 at pH 7.4 when the hydrogel reached an equilibrium state. (**b**) Partial enlarged view of Figure 2a. It shows that the hydrogel swelled faster in the first 1 h at pH 5 and 6. (**c**) Vertical view of three hydrogel bars (same initial size) when they reach an equilibrium state at pH 5, 6 and 7.4 conditions.

**Figure 3 ijms-24-04616-f003:**
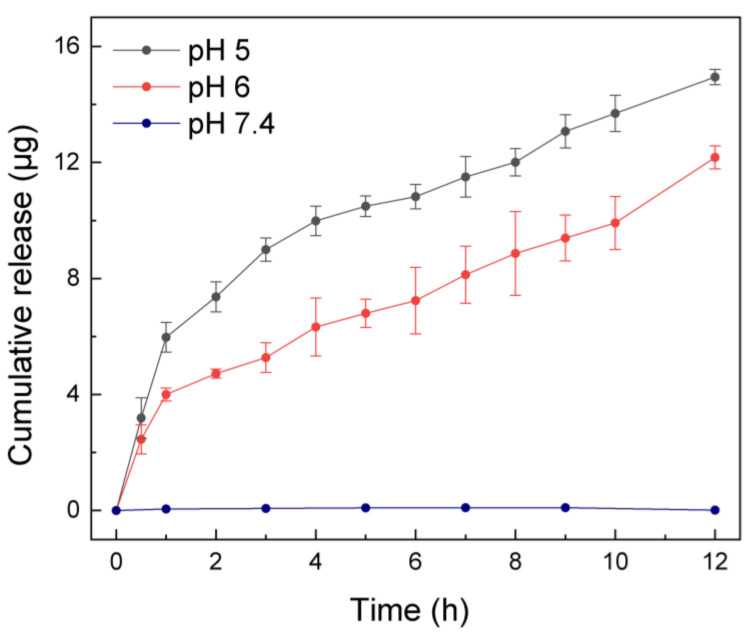
Cumulative release amount by asymmetric microfluidic/chitosan device at three different pH values (5, 6 and 7.4) over 12 h. Minocycline only released at pH 5 (total released amount was 14.94 µg in 12 h) and pH 6 (total released amount was 12.17 µg in 12 h) but not in a neutral environment due to the higher swelling ratio of the hydrogel in acid environments. In the first hour, the average flowrate was 5.97 µg/h (at pH 5) and 3.99 µg/h (at pH 6), then stabilized between 0.51 and 1.63 µg/h (at pH 5) and 0.44 and 1.13 µg/h (at pH 6), respectively.

**Figure 4 ijms-24-04616-f004:**
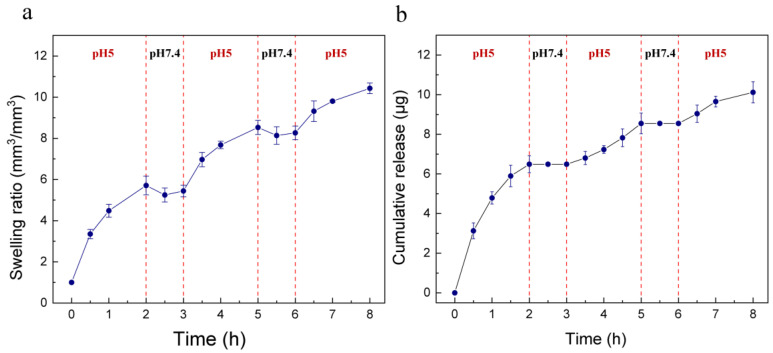
(**a**) The swelling ratio of hydrogel, which was subjected to a low–high pH (5, 7.4) cycle over 8 h. The hydrogel showed de-swelling behavior when it was transferred from pH 5 to a pH 7.4 condition. (**b**) Minocycline release performance of the device at alternate pH values of 5 and 7.4 over 8 h. Minocycline only released at pH 5 and the cumulative release amount was 10.12 µg at the end of 8 h.

**Figure 5 ijms-24-04616-f005:**
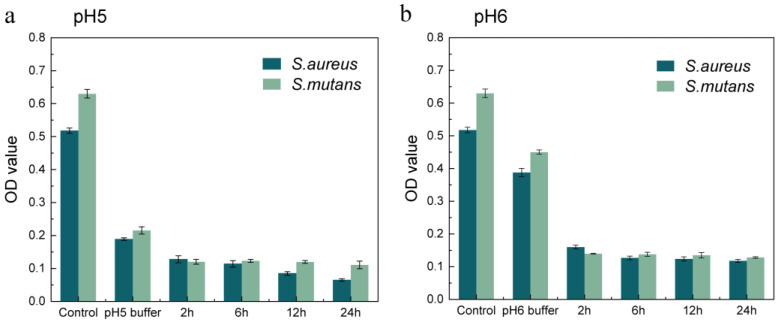
Assessment of the antibacterial ability of minocycline released from the asymmetric microfluidic/chitosan device under pH 5 (**a**) and 6 (**b**) environments. The released solution from each condition was cultured with bacterial suspensions in a 1:1 (*v*/*v*) ratio, and after cocultivation the OD values were measured.

**Figure 6 ijms-24-04616-f006:**
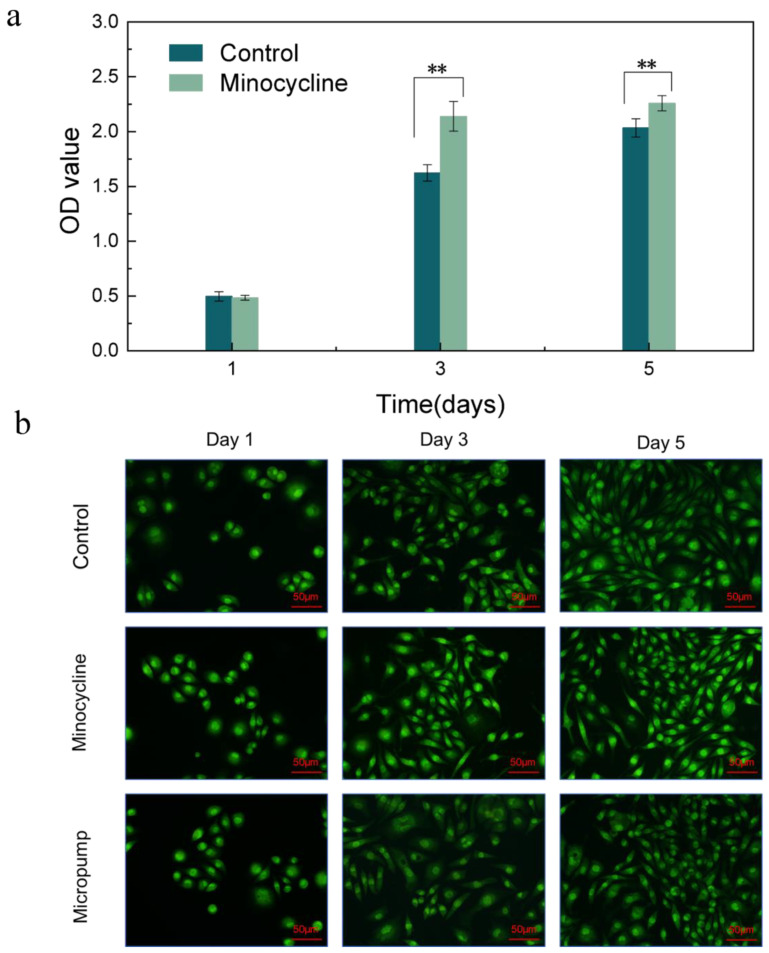
(**a**) CCK-8 assay results of control and minocycline conditions through 5 days. ** *p* < 0.01 (**b**) LSCM images of fibroblasts seeded in wells (Control), seeded in wells with minocycline-containing medium (Minocycline) and seeded on the surface of micropump (Micropump). Scale bar in LSCM images is 50 µm.

**Figure 7 ijms-24-04616-f007:**
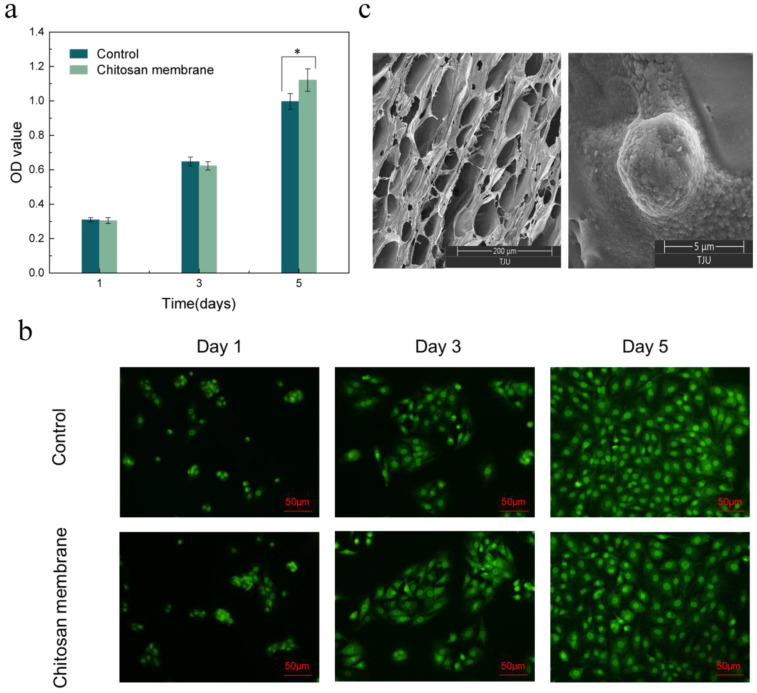
(**a**) CCK-8 assay results of osteoblasts seeded into wells (Control) and on the surface of the chitosan membrane. * *p* < 0.05 (**b**) LSCM images of osteoblasts in wells and attached to the chitosan surface. Scale bar in LSCM images is 50 µm. (**c**) SEM images of the surface of chitosan membrane in 200 µm scale (left), and osteoblasts attached on the chitosan surface after 5 days culturing in 5 µm scale (right).

**Figure 8 ijms-24-04616-f008:**
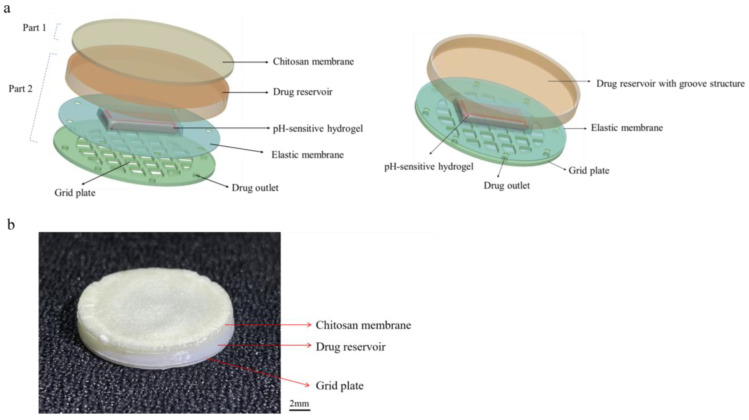
The structure of the asymmetric microfluidic/chitosan device. (**a**) 3D exploded view of the asymmetric microfluidic/chitosan device, from top to bottom: chitosan membrane, drug reservoir, pH-sensitive hydrogel wrapped by an elastic membrane, grid plate. (**b**) Side view of the fabricated asymmetric microfluidic/chitosan device.

**Figure 9 ijms-24-04616-f009:**
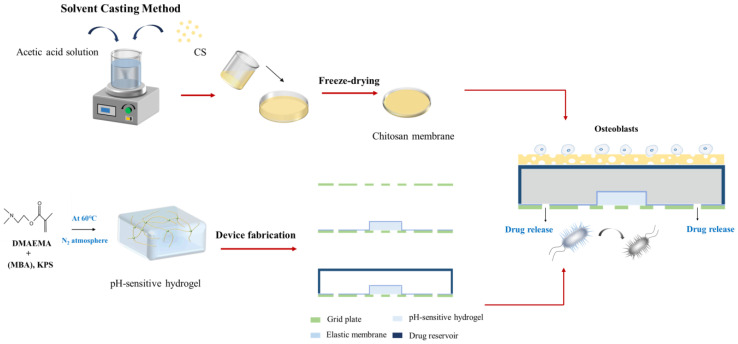
The process of fabricating the asymmetric microfluidic/chitosan device. After the hydrogel was synthesized, it was wrapped by an elastic membrane and embedded in the drug reservoir. The loose chitosan membrane prepared by the solvent casting method was adhered on the upper layer of the drug reservoir.

## Data Availability

Not applicable.

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
