# Peer review of "A pH-Responsive Asymmetric Microfluidic/Chitosan Device for Drug Release in Infective Bone Defect Treatment"

_ijms, 2023, doi:10.3390/ijms24054616_

Round 1
Reviewer 1 Report
Comments to the Author
The investigation of bacterial infection is of practical importance for guided bone regeneration therapy. In this manuscript, the authors provide experimental results of bacterial infection in the common pH range. A pH-responsive drug release asymmetric microfluidic/chitosan device was proposed to act as a promising therapeutic approach in the treatment of infective bone defects. The present topic is interesting and the experiments are well organized. Experimental results could guide the treatment of infective bone defects. Therefore, I am willing to recommend it for publication in IJMS if the comments below can be addressed.
Comments and Suggestions:
(1) Three different pH buffer solutions (pH 5, 6 and 7.4) were tested in the present study, however, the pH range of an infected region is 4.5-6.5 as mentioned in Line 79. Besides, it shows a faster and larger swelling ratio in acid environments in the present study. So, did the author test results for pH 4.5 (a more acid environment) or could the author predict if it would show a significant difference for pH 4.5?
(2) All tests were repeated three times and averaged. However, there are some missing error bars, such as results for pH 7.4 in Figure 5, Ph 6 buffer for S. aureus in Figure 7 (b). Please add explanations or results for repeated experiments.
(3) It’s better to provide the definition of some terminological names, such as OD (Line 355), p (Line 263). Please check the whole manuscript.
(4) In section 2.6, 1:1 (v/v) ratios are chosen for all experiments. Were they similar to the most common infection environments? Will different ratios affect experimental results?
(5) For Figure 4(b), the X-axis has a more accurate label compared with Figure 4(a). So, it’s better to add more number labels for Y-axis with the same accuracy as the X-axis (two decimal places) to make readers clearer.
(6) There are some typos and grammar mistakes. For example, for Line 356, “buffer release solution” should be revised to “buffer release solutions”; For Lines 241-242, there should be a space between the number and their units, such as “100 mg/mL” and “100 U/mL”. Please check the whole manuscript and make corrections.
Reviewer 2 Report
The manuscript deciphers a novel study and the proposed hypothesis is well achieved by adopted methodology. The results are clear and easy to understand.
Major issues are:
1. Authors must improve the language of the manuscript.
2. In the introduction section the authors should highlight the need for this new study and its impact on solving the health issues with the help of prior arts.
3. The discussion part is missing. Authors should discuss the merit of current study based on the superiority of their obtained results by providing a comparison with previous works carried out in this area.
Reviewer 3 Report
1. Provide scale bar for fig. 8b and 9b
2. Add digital images showing the swelling of the hydrogel at different pH
3. Elemental mapping on Fig. 9C
Round 2
Reviewer 3 Report
The authors have addressed all my comments.